3

5

# Assessing Spacing Impact on the Wind Turbine Array Boundary Layer via Proper Orthogonal Decomposition

Naseem Ali, Nicholas Hamilton, and Raúl Bayoán Cal

Department of Mechanical and Materials Engineering,

Portland State University, Portland, OR 97207

# Abstract

A 4 x 3 array of wind turbines was assembled in a wind tunnel with four cases to study the influence based on streamwise and spanwise spacings. Data are extracted using stereo particleimage velocimetry and analyzed statistically. The maximum mean velocity is displayed at the upstream of the turbine with the spacing of 6D and 3D, in streamwise and spanwise direction, respectively. The region of interest downstream to the turbine confirms a notable influence of the streamwise spacing is shown when the spanwise spacing equals to 3D. Thus the significant impact of the spanwise spacing is observed when the streamwise spacing equals to 3D. Streamwise averaging is performed after shifting the upstream windows toward the downstream flow. The largest and smallest averaged Reynolds stress, and flux locates at cases  $3D \ge 3D$  and  $6D \ge 1.5D$ , respectively. Snapshot proper orthogonal decomposition is employed to identify the flow coherence depending on the turbulent kinetic energy content. The case of spacing  $6D \ge 1.5D$  possesses highest energy content in the first mode compared with other cases. The impact of the streamwise and spanwise spacings in power produce is quantified, where the maximum power is found in the spacing of  $6D \ge 3D$ .

6 PACS numbers:

# 7 I. INTRODUCTION

Impacts on siting wind turbines in the wind farm include interaction between wakes, 8 decreased wind velocity and an increased coalesce dynamic load on the downwind turbines. q Turbine wakes lead to loss an average 10-20% of the total potential power output [1]. Ex-10 tensive experimental and numerical studies focused on the wake properties in terms of mean 11 flow characteristics and the specifications of the turbulent flow utilized to obtain optimal 12 ower production. Wake growth particularly depends on the shape and magnitude of the 13 velocity deficit that relies on the ground roughness, flow above the canopy, and spacing 14 between the turbines. 15

Chamorro and Porté-Agel [2] studied the influence of surface roughness on the wake as 16 it alters the velocity distribution as well as turbulence levels. Cal et al. [3] noticed that the 17 order of magnitude of kinetic energy entrainment is nearly equal to the power harvested by 18 the wind turbine. Calaf et al. [4] used large Eddy simulation (LES) model to determine 19 the roughness length scale of the fully developed wind turbine array boundary layer and 20 quantified the impact of the correlation between the mean flow and turbulence. Meyers 21 and Meneveau [5] compared aligned versus staggered wind farms; the latter yielding a 5% 22 increase in extracted power. Chamorro and Porté-Agel [6] examined the wind farm under 23 neutral stratification, observing flow can be divided into two regions that develop at different 24 rates. The first region is located below the top tip and reaches the fully developed condition 25 after the third row of turbines. The second region is located above the top tip where the 26 flow modifies slowly. Hamilton et al. [7] investigated the effect of wind turbine configuration 27 on the wake interaction and canopy layer. They considered standard Cartesian and row-28 offset configurations. The results showed that the maximum flux of kinetic energy increases 29 about 7.5% in the exit row of offset configuration compared with the Cartesian arrangement. 30 Hamilton et al. [8] studied the anisotropy of the Reynolds stress in the wake of wind turbine 31 arrays in for counter-rotating turbines. The result showed that the greater magnitude of the 32 flux can be entrained when the rotation direction of the blades is changed in a row-by-row 33 configuration. 34

Although there are many studies dealing with the effect of the density of turbines on the wake recovery, it is still a debated question. The optimal spacing according to the Nysted farm is 10.5 diameters (D) downstream by 5.8D spanwise, whereas according to the Horns

Rev farm is 7D, optimal spacing along the bulk flow direction and 9.4D or 10.4D along 38 the diagonal. Barthelmie and Jensen [9] showed that the spacing in the Nysted farm is 39 responsible for 68-76% of the farm efficiency variation and for wind speed below 15 ms<sup>-1</sup>, 40 the efficiency will increase 1.3% for every one diameter increasing in spacing. Hansen et al. 41 [10] pointed out that the variations in the power deficit for different spacing were almost 42 negligible at approximately 10D into Horns Rev farm in spite of a large power deficit resulting 43 from smaller turbine spacing. In addition, the mean power deficit is similar along single wind 44 turbine rows when inflow direction is unified and the wind speed interval from 6 to  $10 \text{ ms}^{-1}$ . 45 Furthermore, the maximum deficit happens between the first and the second row of turbines 46 and minimum deficit in the remaining downstream. González-Longatt [11] found that when 47 the downstream and spanwise spacing increased, the wake coefficient representing the total 48 power output with wake effect over total power without wake effect increased, and the effect 49 of the incoming flow direction on the wake coefficient increased when the spacing of the 50 array is reduced. Meyers and Meneveau [12] studied the optimal spacing in fully developed 51 wind farm with considerable limitations including neutral stratification and flat terrain with 52 no topography. The results highlight that depending on the ratio of land cost and turbine 53 cost, the optimal spacing might be 15D instead of 7D. Stevens [13] used the effective 54 roughness length performed by LES to predict the wind velocity at hub height depending 55 on the streamwise and spanwise spacing, and the turbine loading factors. Also showing that 56 optimal spacing depends on the wind farm length in addition to the factors suggested in 57 [12]. Stevens et al. [14] used LES model to investigate the power output and wake effects in 58 aligned and staggered wind farms with different streamwise and spanwise turbine spacing. 59 In the staggered configuration, power output in fully developed flow depends mainly on 60 the spanwise and streamwise spacings, whereas in the aligned configuration, power strongly 61 depends on the streamwise spacing. 62

In this article, the proper orthogonal decomposition (POD) analysis will be employed to identify the structure of the turbulent wake associated with variation in spacing and understand the effect of the streamwise and spanwise on the characteristic flow of the wind turbine array, including Reynolds shear stress, turbulent flux and energy production.

# 67 II. SNAPSHOT PROPER ORTHOGONAL DECOMPOSITION

Balancing between the gain and loss in energy can be quantified through the mean kinetic 68 energy equation [15]. One of the main gain sources can be obtained by the spatial transport 69 of energy by Reynolds shear stress, named the energy flux. The Reynolds shear stress is the 70 center of the energy flux, therefore this study will focus on the energy flux to quantify the 71 impact of the streamwise and spanwise spacing through the statistical analysis and using 72 Proper orthogonal decomposition. POD is a mathematical tool that depends on a set of F 73 snapshots to obtain the optimal basis functions and decompose the flow into modes that 74 express the most dominant features. This technique, which is presented in the frame of 75 turbulence by Lumely [16], categorizes structures within the turbulent flow depending on 76 their energy content and allows for filtering the structures associated with the low energy 77 level. Sirovich [17] presented the snapshot POD that relaxes the difficulties of the classical 78 orthogonal decomposition. 79

The flow field, taken as the fluctuating velocity, can be represented as  $u_i = u(\vec{x}, t^n)$ , where  $\vec{x}$  and  $t^n$  refer to the spatial coordinates and time at sample n, respectively. A set of the orthonormal basis functions,  $\phi$ , can be presented as

$$\phi_i = \sum_{n=1}^{N} A(t^n) u(\vec{x}, t^n).$$
(1)

The optimal functions have minimum averaged error and maximum averaged projection in
mean square sense. The largest projection can be determined using the two point correlation
tensor and Fredholm integral equation

$$\int_{\Omega} R(\vec{x}, \vec{x}') \phi(x') dx' = \lambda \phi(x), \tag{2}$$

where  $R(\vec{x}, \vec{x}')$  is a spatial correlation between two points  $\vec{x}$  and  $\vec{x}'$ , N is the number of snapshots, T is the transpose of the matrix, and  $\lambda$  are the eigenvalues. The optimal deterministic problem is solved numerically as the eigenvalue problem. The eigenfunctions are orthogonal and have a corresponding positive and real eigenvalues organized by descending arrangement. The POD eigenvectors illustrate the spatial structure of the turbulent flow and the eigenvalues measure the energy associated with corresponding eigenvectors. The summation of the eigenvalues presents the total turbulent kinetic energy (E) in the flow

 $_{93}$  domain. The fraction of the cumulative energy,  $\eta$  and the normalized energy content of each

<sup>94</sup> mode,  $\xi$ , can be represented as,

$$\eta_n = \frac{\sum_{n=1}^n \lambda_n}{\sum_{n=1}^N \lambda_n},\tag{3}$$

$$\xi_n = \frac{\lambda_n}{\sum_{n=1}^N \lambda_n}.$$
(4)

POD tool is particularly useful in rebuilding the Reynolds shear stress using a set of eigen functions as follows,

$$\langle u_i u_j \rangle = \sum_{n=1}^N \lambda_n \phi_i^n \phi_j^n.$$
(5)

POD used to describe coherent structures of different types of flow such that axisymmetric 97 mixing layer [18], channel flow [19], atmospheric boundary layer [20], wake behind disk [21], 98 and subsonic jet [22]. In the frame of a wind turbine wake flow, Anderson *et al.* [23] applied 99 POD to the flow in a wind farm simulated using LES. They showed the large scale motion 100 and dynamic wake meandering are strongly governed by turbine spacing. The number of 101 modes required to reconstruct the flow is related to the flow homogeneity. Hamilton et 102 al. [24] investigated the wake interaction and recovery dynamic for Cartesian and row-103 offset wind array, showing that the flux of turbulence kinetic energy are reconstructed with 104 approximately 1% of the total modes. Bastine *et al.* [25] performed analysis for a single wind 105 turbine modeled via LES, observing the three modes is sufficient to capture the dynamic of 106 the effective velocity over a potential disk. Recently, VerHulst and Meneveau [26] applied 107 three dimensional POD on the LES data and quantified the contribution of each POD mode 108 to the energy entrainment, finding that the net entrainment is relevant to the layout of the 109 wind turbines in the field. 110

#### 111 III. EXPERIMENTAL DESIGN

A 4 x 3 array of wind turbines was placed in the closed- circuit wind tunnel at Portland State University to study the effects due to variation in streamwise and spanwise spacing in a wind turbine array. The dimensions of the wind tunnel test section are 5 m (long), 1.2 m (wide) and 0.8 m (height). The entrance of the test section is conditioned by the passive

FIG. 1: Experimental Setup. Dashed gray lines indicate the placement of the laser sheet relative to the model wind turbine array. Filled gray boxes indicate measurement locations discussed below. Figure reproduced from Hamilton *et al.* [8].

grid, which consists of 7 horizontal and 6 vertical rods, to introduce large-scale turbulence.
Nine vertical Plexiglas strakes located at 0.25 m downstream of the passive grid and 2.15 m
upstream the first row of the wind turbine were used to modify the inflow. The thickness
of the strakes is 0.0125 m with a spanwise spacing of 0.136 m. Surface roughness elements
were placed on the wall as a series of chains with diameter of 0.0075 m and spaced 0.11 m
apart. Figure 1 shows the schematic of experimental setup.

A 0.0005 m thick steel was used to construct 3 bladed wind turbine rotor. The diameter of 122 the rotor was 0.12 m, equal to the height of the turbine tower. Each rotor blade was pitched 123 at  $15^{\circ}$  out of plane at the root and  $5^{\circ}$  at the tip. These angles were chosen to provide angular 124 velocity that correlates with required ranges of tip-speed ratio. A DC electrical motor of 125 0.0013 m diameter and 0.0312 m long formed the nacelle of the turbine and was aligned with 126 flow direction. A torque sensing system was connected to the DC motor shaft following the 127 design outlined in [27]. Torque sensor consists of a strain gauge, Wheatstone bridge and the 128 Data Acquisition with measuring software to collect the data. For more information on the 129 experiment conditions and data processing, see [7]. 130

In this study, the flow field was sampled in four configurations of a model-scale wind turbine array, classified as  $\Pi_n$ , where n varies from 1 through 4 and considered in Table I. Permutations of streamwise spacing of 6D and 3D, and spanwise spacing of 3D and 1.5D are examined. Stereoscopic particle image velocimetry (SPIV) was used to measure streamwise, wall-normal and spanwise instantaneous velocity at the upstream and downstream of the wind turbine at the center line of the fourth row as shown in figure 2. At each measurement