# Peer review of "Assessing Spacing Impact on the Wind Turbine Array Boundary Layer via Proper Orthogonal Decomposition"

_Wind Energy Science, 2016_

## Referee Comment (RC1) · Anonymous Referee #1 · 26 Jul 2016

This manuscript focuses on POD analysis of Stereo-PIV measurements performed upstream and downstream of the most downstream turbine in a 4 x 3 array. Four different layouts were investigated by varying two streamwise and spanwise separation distances.

Introduction is a bit unfocused and writing can be improved. From the abstract and the introduction, the scientific goal of this manuscript is unclear and, frankly speaking, it is still unclear even at the end of the document. Maybe is the reconstruction of the Reynolds stress through POD modes and then highlighting difference among the different layouts? This should be explained clearly.

In Sect. 2, POD is briefly described, not always in a very rigorous manner (see detail

comments below). In Sect. 3 the experimental setup is described. In Sect. 4 power production for the different layouts and rotor angular velocities is reported. This section looks a bit disconnected with the remainder of the manuscript. No clear connections are made between power production and the wake velocity field. In Sect. 5A, mean velocity field and Reynolds stress are reported for the different layouts. In Sect. 5B, streamwise averaging analysis is described. In Sect. 5C, the POD results are provided, while in Sect. 5D, reconstruction of the Reynolds stress through POD modes is presented.

The main criticism on this manuscript is that the results presented do not provide a clear insight to improve understanding of wake turbulence for different wind farm layouts. I guess the main results should be better highlighted in the data analysis and conclusions. Furthermore, writing should be improved throughout the manuscript.

**Detail comments**

1. Abstract: possesses highest energy content, it sounds strange. Consider to rephrase it.

2. line 9: increased coalesce dynamic load. Consider to rephrase it.

3. line 13: Wake growth particularly depends on the shape and magnitude of the velocity deficit. What does it mean?

4. line 36: according to the Nested farm..Spacing within a real wind farm varies with wind direction. I suggest providing ranges of spacing according to the wind rose.

5. line 68: Maybe it is appropriate to write the mean kinetic energy budget?

6. line 70: The Reynolds shear stress is the center of the energy flux. This is not clear, consider to rephrase it.
7. lines 78-79: The snapshot method enables reducing POD computational cost when the space dimension of a single snapshot is larger than the total number of snapshots.

8. line 88: The optimal deterministic problem is solved numerically as the eigenvalue problem. Rephrase it.

9. Fig. 3: This is the power measured only for the turbine under examination or from the entire array? Is it possible to report the power in a non-dimensional form?

10. Sect. 4: The power measurements seem to be disconnected from the rest of the manuscript. Try to bridge the power measurements with the wake velocity data.

11. line 161: do you mean upstream and downstream of the turbine under examination?

12. line 186: there is an extra space.

13. Fig. 4: why you do not show the other two velocities as well?

14. Sect. 5B: explain better the rationale in performing averaging in the streamwise direction even though streamwise gradients are significant.

15. line 254: specify if POD was performed by analyzing snapshots of the Reynolds stress or velocity components.

---

## Author Comment (AC1) · 9 Aug 2016

The comment was uploaded in the form of a supplement:
http://www.wind-energ-sci-discuss.net/wes-2016-23/wes-2016-23-AC1-supplement.zip

---

## Referee Comment (RC2) · Anonymous Referee #2 · 10 Aug 2016

Review of "Assessing Spacing Impact on the Wind Turbine Array Boundary Layer via Proper Orthogonal Decomposition" by Ali et al.

The authors use arrays of wind turbines in a wind tunnel to study the effect of streamwise and span-wise spacing on the wind turbine boundary layer. They test four different arrangements of twelve turbines, examine the resulting power production, velocity deficits upwind and downwind, as well as Reynolds stresses and kinetic energy flux. They then apply POD to these fields and describe the differences, and create reconstructions of these modes. The differences between the four cases are moderately interesting, but unfortunately the manuscript as it is currently written does not clearly define any unique contribution of this work to the literature.

[Figure]

Although it's clear that a lot of work went into this paper and the wind tunnel study behind it, substantial effort must be applied to it to make this manuscript suitable for publication. The scientific goal, or hypothesis, or driving question must be presented clearly (and it is not in the current form). The community understands the benefit of larger stream-wise spacing – is the goal here to assess the role of cross-stream spacing? Further, much text is devoted to describing the results of the POD and the differences between modes, but unfortunately it's not clear what new knowledge or insight is obtained from the POD analysis. What would a reader learn from this study that he/she did not know before?

Further, the authors should remember that one of their goals is to make their results as clear as possible to the reader. In its current form, the paper is very difficult to read and understand. The senior authors should provide a much more careful review of the writing style. Many sentences are confusing, even in the abstract (which should provide a very clear and concise summary of the paper – no one will read the paper if the abstract is confusing). For example: "The region of interest downstream to the turbine confirms a notable influence of the streamwise spacing is shown when the spanwise spacing equals to 3D." What is the subject in this sentence? What is the verb? Please try to make the sentences as short and simple as possible to ensure they are more clear. Unless the writing is revised carefully, I cannot see that this paper would be appropriate for publication.

Just as many individual sentences are very confusing, the overall structure of the paper is also confusing. For example, why does "Power Measurements" get a heading while everything else is folded into the "Results"? The power measurements should become part of the discussion of the streamwise velocity.

Specific major comments:

1. A clear hypothesis must be stated, and the value of the POD must be stated explicitly.

2. The figures are not designed intuitively. Although four test cases are examined

repeatedly, they are given names with no correlation to what they stand for. I under-
stand the appeal of brief labels for the cases – it's more convenient for writing – but
it's also more confusing for the reader. Perhaps labels like 6X3, 3X3, 3X1.5, 6X1.5
would facilitate the interpretation of the images? Similarly, wouldn't it be more intuitive
to have top left 6X3, top right 3X3, bottom left 6X1.5, bottom right 3X1.5? In this fash-
ion, the rows are organized according to the span-wise spacing and the columns are
organized according to the stream-wise spacing, which makes it easier for the reader
to do comparisons between the cases.

3. Please try to start each paragraph with a topic sentence. For example, Line 16 jumps
into a literature review, and the reader is not sure what the point is. Of the numerous
wind tunnel studies and LES (many of which have been omitted from this literature
review) studying wind farms, why are these studies the important ones in reference of
this particular study? This is just an example of many cases where paragraphs jump
into a description of this or that figure without indicating to the reader what the point is
of the discussion.

4. Speaking of the literature review, numerous other LES of wind farms
(http://www.nrel.gov/docs/fy12osti/53554.pdf, among others) have been presented in
the literature. What is the justification for omitting them?

5. Is there any thermal forcing in these cases? This should be mentioned.

6. The upwind stream-wise velocity contours for cases 2 and 3 seem very surprising.
If this decrease of velocity is due to an induction zone in front of the farm, shouldn't
the lower velocities be closer to the turbines (ie at x=-1 D) rather than further away (at
x=-1.8D)? The discussion in line 164 should explain this odd phenomena rather than
just describe it.

7. All the velocities (Figure 4) should be normalized with respect to the desired inflow
velocity at hub height.

8. The motivation for the extensive POD discussion is never presented. What have we learned from the POD that we did not know before? It is not enough to state that "The findings of this study have a number of practical implications" without stating what those implications are directly.

Specific minor comments:

1. The abstract is organized in a confusing fashion: please put all the set-up information first, and then the results. Mixing them together ("Streamwise averaging. . ..") appears after some of the results

2. Line 36: "optimal" is not the appropriate word here. "actual" makes more sense – the wind farm designers were considering many variables when constructing those wind farms.

3. lines 35-62: please break up this paragraph: the first idea is about density for aligned wind farms, then at some point staggered design is introduced. That should get its own paragraph (if it is important).

4. lines 63-66. Very abrupt transition to POD. It, and its use in wind energy research, should be introduced.

5. lines 63-66: Please provide a few sentences outlining the structure of the paper.

6. line 79: isn't POD widely used in wind energy? Shouldn't some of those papers be cited here? (I see now that I have read down to 97 that a short review is presented there, but it should come earlier in the paper.)

7. Figure 1: has the publishing company of Hamilton et al. given permission for the figure to be reproduced here?

8. line 145-146: how were erroneous field identified? How many were there? Does this undermine the reader's confidence in the measurements?

9. Somewhere in the discussion of Table 1 it should be pointed out that no staggered

grids were considered.

10. Table 1/Figure 3/Figure 7: I understand why you might want to use brief labels for the cases, but can you choose labels somewhat more clear, like 6X3, 3X3, 3X1.5, 6X1.5 to facilitate the interpretation of the images?

11. Figure 4, 5, 6: Please use small letters to clearly define what each panel is showing. (Thank you for using a clear color table.) Wouldn't it be more intuitive to have top left 6X3, top right 3X3, bottom left 6X1.5, bottom right 3X1.5? Also consider overlaying a contour level at some important threshold.

12. line 174: please summarize, providing a ranking of the cases corresponding to their spacings

13. In the conclusion, please first redefine the cases before describing their results.

Particularly confusing sentences: Please review the entire manuscript carefully to ensure coherency and correct English grammar. In many places the intent of the writing is muddied by the composition of the sentences. Some of these are noted below, but the entire manuscript should be reviewed.

1. abstract: "The region of interest downstream...."

2. abstract: "The impact of the streamwise....in power produce..." should be produced perhaps?

3. line 9: coalesce?

4. line 10-13: where is the verb?

5. many between 13 and 65 ....

6. line 65: missing a word

7. line 68: "Balancing" should be "The balance"

8. line 71: "center" ?

9. line 76: misspelling of Lumley. Using bibliographic software can reduce errors like this.

10. line 89

11. line 118

12. line 122

13. line 132

14. line 137-140

15. Table I: is "spacing area" the correct term for this?

16. line 155: majority? Do you mean maximum?

17. 159-160

18. 173-174

19. 178-179

20. please carefully review the rest of the manuscript

---

## Author Comment (AC2) · 25 Aug 2016

The comment was uploaded in the form of a supplement:
http://www.wind-energ-sci-discuss.net/wes-2016-23/wes-2016-23-AC2-supplement.zip
* * *